# Hybrid Framework for Diabetic Retinopathy Stage Measurement Using Convolutional Neural Network and a Fuzzy Rules Inference System

Rawan Ghnemat 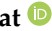

Computer Science Department, King Hussein Faculty for Computing Sciences, Princess Sumaya University for Technology, P.O. Box 1438, Amman 11941, Jordan; r.ghnemat@psut.edu.jo

**Abstract:** Diabetic retinopathy (DR) is an increasingly common eye disorder that gradually damages the retina. Identification at the early stage can significantly reduce the severity of vision loss. Deep learning techniques provide detection for retinal images based on data size and quality, as the error rate increases with low-quality images and unbalanced data classes. This paper proposes a hybrid intelligent framework of a conventional neural network and a fuzzy inference system to measure the stages of DR automatically, Diabetic Retinopathy Stage Measurement using Conventional Neural Network and Fuzzy Inference System (DRSM-CNNFIS). The fuzzy inference used human experts' rules to overcome data dependency problems. At first, the Conventional Neural Network (CNN) model was used for feature extraction, and then fuzzy rules were used to measure diabetic retinopathy stage percentage. The framework is trained using images from Kaggle datasets (Diabetic Retinopathy Detection, 2022). The efficacy of this framework outperformed the other models with regard to accuracy, macro average precision, macro average recall, and macro average F1 score: 0.9281, 0.7142, 0.7753, and 0.7301, respectively. The evaluation results indicate that the proposed framework, without any segmentation process, has a similar performance for all the classes, while the other classification models (Dense-Net-201, Inception-ResNet ResNet-50, Xception, and Ensemble methods) have different levels of performance for each class classification.

**Keywords:** diabetic retinopathy (DR); computer vision; automation; convolutional neural network (CNN); fuzzy inference system (FIS); transfer learning

## 1. Introduction

Diabetic retinopathy (DR) is a major cause of vision loss worldwide [1]. Regular screening for DR can detect early features or symptoms. However, human experts in this domain still perform the diagnosis. Computer-aided disease diagnosis in retinal image analysis could provide a reasonable solution for the screening and diagnosis processes. Automated tools for clinical stage measurements of retinal problems can provide continuous and accurate monitoring of the disease. Advances in artificial intelligence and machine learning approaches enable such application for clinical practice [2].

Recently, an important need has arisen for the automation of an accurate DR detection system, as providing an affordable, accurate system will overcome the problem of a lack of retina specialists around the word [3]. The detection of different patterns in retinal images is a key factor in DR measurement. Retinal tissue in diabetic patients deforms in different ways, such as microaneurysms that appear as tiny red dots on images of early-stage DR. Microaneurysms usually grow into retinal hemorrhages in moderate DR cases, and in some cases yellow or white exudates can observed, while in severe cases, blood vessels leak into the retina, known as macular edema, causing blurry vision [4].

Intelligent health care application has an important function in retinal image analysis and feature extraction, and provides a reasonable solution to control the growth of DR, especially if the detection occurs at the early stages; moreover, deep learning methods for DR

grading have achieved significantly improved performance [5]. However, accurate classification for DR grading remains challenging due to many reasons, such as the insufficiency of training samples and the poor quality of fundus images taken using different devices [6]. The measurement of the DR severity stage is a difficult task due to the differences in the sizes of lesions among fundus images of the same class and visual similarities in detected features, such as shapes and colors between the fundus images of different classes [7].

Hybrid artificial intelligence approaches face performance challenges in CNN models. Zhang et al. [8] proposed a Hybrid Graph Convolutional Network (HGCN) for diabetic retinopathy grading with limited labeled data and a large amount of unlabeled data (semi-supervised learning), and the experimental results showed the better performance of HGCN in semi-supervised retinal image classification.

Considering all of these facts, the proposed work implemented a simple CNN for feature extraction and combined it with a rule-based expert system for better classification using the model used in [9]; the model was trained using Kaggle public datasets [10], and the rules changed based on the new domain of application. The grading of diabetic retinopathy was implemented without segmentation; furthermore, the framework gives similar performance for all classes. This study provides important contributions to this field by enhancing the accuracy and the performance based on a hybrid approach and a relatively small amount of data.

The goal of the proposed hybrid intelligent framework consisting of a conventional neural network and a fuzzy inference system to efficiently measure the stages of DR.

- The framework has similar performance for all classes, overcoming the problem of different data sizes in each training class;
- The framework does not need a segmentation phase;
- The framework adds a rule-based system based on human experts' knowledge to the deep learning model;
- The evaluation and comparison with related models show that the framework outperforms the other models.

The rest of this paper is organized as follows: Section 2 provides the background and previous works. In Section 3, we present the research methodology in detail. Section 4 shows the evaluation and experimental results. Finally, in Section 5, we conclude our work and identify future work avenues.

## 2. Background and Previous Work

Diabetic retinopathy (DR) appears in people with a medical history of diabetes [11] and high blood glucose levels. Many researchers have worked on DR symptom detection using feature recognition techniques [12].

Other methods are based on recognizing the retinal blood vessels and pathologies from fundus images as features and classifying the diabetic retinopathy severity grades [13]. Feature extraction and image analysis for DR classification show great potential for DR grading; however, they excessively depend on labelled data. These methods rely on pixel-level annotation data. This type of annotation is useful in techniques to locate lesions within an image and segment out regions of interest from the background [14].

Image processing techniques using machine-learning methods suffer from the lack of domain experts for validation [15]; in such cases, the dependency on data and statistical models without human experts' validation is still questionable [16].

Currently, deep learning and convolutional neural networks (CNNs) are frequently used in medical applications with computer vision, especially the automated detection of diabetic retinopathy [17,18]. Extracting important features, such as hard exudates, blood vessels, and texture [19], using a transfer learning-based CNN architecture from fundus images performs relatively well.

CNN studies address the grading of non-proliferative DR categories, namely mild, moderate, and severe stages, using a transfer learning-based DR detection system [20], but

performance issues face most of the deep learning methods due to the small number of DR fundus images used to train a deep CNN model; hence, overfitting problems appear [21–24].

Convolutional neural networks (CNNs) are a powerful tool for DR detection, which includes different tasks: classification [25], segmentation [26], and detection [27].

Researchers [28] have coupled CNNs with transfer learning and hyper-parameter tuning, adopted AlexNet, VggNet, GoogleNet, and ResNet, and analyzed how well these models handle DR image classification, using Kaggle datasets to train these models. The best classification accuracy is 95.68% using transfer learning with data augmentation, where the fundus images data were increased to 20 times the original.

The authors of [29] (Resnet50, Inceptionv3, Xception, Dense121, Dense169) enhanced the classification of different stages of DR. The experimental results show that the proposed model detects all of the stages of DR. The achieved accuracy is 80.8%.

Many hybrid methods based on CNN and other intelligent methods have been proposed, such as the Swarm Optimization (PSO) algorithm-based Convolutional Neural Network (CNN) Model, also called the PSO-CNN model [30], to detect DR from color fundus images. Many proposed hybrid CNN models used preprocessing, feature extraction, and classification [31].

Orujov et al. [32] used feature extraction for blood vessels in retinal fundus images using a contour detection algorithm based on fuzzy rules. Fuzzy rules were applied to image gradient values to extract edges and make DR classification decisions based on membership functions. The results of this model offered a similar performance to CNN methods, but it contains flexible rules, offering an alternative to current deep learning applications, severity classification of DR using CNN and attention module proposed in [33], which reduced both the complexity of the model and the training time needed.

This research work proposed a method for fine-tuning a pre-trained CNN model for DR grading using fuzzy rules and fundus images. The method takes a retinal fundus image as the input, the CNN model processes it with the fine-tuned model and grades it into normal or DR levels and then the fuzzy system takes the processed images and classifies them based on human experts' rules into four categories (normal, mild, moderate and severe) with the grading percentage.

An intelligent computer-aided diagnosis framework for the DR grading of retinal fundus images is implemented, and the framework does not need any segmentation process for the retinal fundus images.

The proposed framework has two parts. The first part embeds the DR lesion structures in a pre-trained CNN model. The second part uses the extracted features of retinal fundus images in a fuzzy inference system (FIS), which significantly reduces the model complexity and data dependency and measures the severity percentage, so multiple uses of the system can provide the progression rate in this case.

Table 1 summarizes the related works' techniques, findings, and limitations compared to the proposed framework.

**Table 1.** Comparison of studies conducted with diabetic retinopathy image datasets and the proposed study.

| Model | References | Data Used | Performance (Accuracy) | Limitation |
|---|---|---|---|---|
| DenseNet | [33] | APTOS dataset (https://www.kaggle.com/c/aptos2019-blindness-detection, accessed on 12 October 2022) | 0.9580 | Model implementation used small and imbalanced datasets. |
| Inception-ResNet | [24] | Customized dataset. | 91.61 | The non-proliferate symptoms are not visible on retina images. Small dataset. |
| ResNet-50 | [34] | Messidor EyePACS | The accuracy achieved ranges from 96%, on a two-category Messidor-2 dataset, to 75.09% on a five-category EyePACS dataset and | The model is highly demanding. Repositories of a large dataset for deep learning. |
| Xception | [35] | IDRiD, | 84% on the binary classification of IDRiD. | Shortage in performance. |

**Table 1.** *Cont.*

| Model | References | Data Used | Performance (Accuracy) | Limitation |
|---|---|---|---|---|
| Ensemble methods | [36] | (APTOS 2019 BD) dataset. | Accuracy of 94.20%. | Noisy images, duplicate images with improper labelling, uneven image resolution, and varying class sample sizes. |
| AlexNet, VggNet, GoogleNet, ResNet, | [28] | Kaggle fundus images data were increased to 20 times the original. | The best classification accuracy is 95.68%. | Huge training dataset. |
| Resnet50, Inceptionv3, Xception, Dense121, Dense169 Ensemble | [29] | Same Kaggle dataset. | The best classification accuracy is 80.8%. | Very long training time. |
| DRSM-CNNFIS | — | Kaggle dataset increased to 3.6 times the original. | The best classification accuracy is 92.8%. | Need for a human domain expert for rule and framework fine-tuning. |

## 3. (DRSM-CNNFIS) Framework

This section presents the steps of the Diabetic Retinopathy Stage Measurement using the Conventional Neural Network and Fuzzy inference system (DRSM-CNNFIS) framework and the dataset used in training and testing.

### 3.1. Dataset Description

Different retinal image types and qualities are available for developing and testing digital screening for diabetic retinopathy. The public dataset used in this work was Kaggle Diabetic Retinopathy Detection [10], which is sponsored by the California Healthcare Foundation.

The Kaggle DR Dataset has 35,126 fundus images for training. Different devices collected the images. Kaggle DR is one of the largest publicly available DR classification datasets; most of the labelling was performed manually and the quality of the images is not homogenous.

These images were labeled using a scale of 0 to 4 based on the severity of diabetic retinopathy (DR). Table 2 shows the five classes of DR as well as their respective percentage from the total data. According to the international clinical diabetic retinopathy scale [35], binary classification, images with labels of 0 and 1 were classified as "No PDR", and relabeled with 0, and images with labels of 2, 3, and 4 were classified as "RDR" and relabeled with 1, as shown in Table 3. The distribution of labels was: {0:25,810, 1:2443, 2:5292, 4:708, 3:873}. Total images: 35,126.

**Table 2.** Distribution of multiclass classification.

| Label | Class | Number of Samples | Percentage |
|---|---|---|---|
| 0 | Normal | 25,810 | 73.84% |
| 1 | Mild NPDR | 2443 | 6.96% |
| 2 | Moderate NPDR | 5292 | 15.07% |
| 3 | Sever NPDR | 873 | 2.43% |
| 4 | Proliferative DR | 708 | 2.01% |

**Table 3.** Distribution of binary classification.

| Label | Class | Number of Samples | Percentage |
|---|---|---|---|
| 0 | No PDR | 28,253 | 80.4% |
| 1 | RDR | 6873 | 19.6% |

Table 2 provides the five grades of the dataset with their percentage, and Table 3 provides the distribution of the binary classification in the training set of the Kaggle dataset. Figure 1 shows sample gradable images from Kaggle DR.

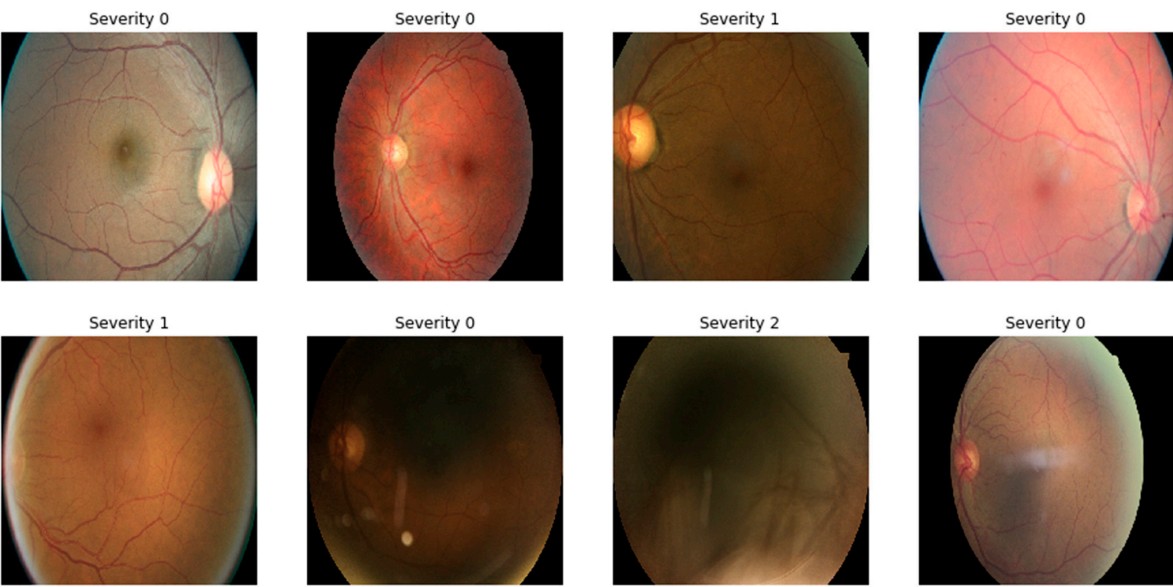

**Figure 1.** Sample gradable images from Kaggle DR.

In this research, the class imbalance issue was addressed using two methods—first, data augmentation, as explained in Section 3.2, and human experts' rules for the results, as explained in Section 3.3.

### 3.2. Image Pre-Processing and Data Augmentation

For preprocessing, first, median filters were used for noise removal and contrast improvement. In addition, images were resized to a standard size of 256 × 256, followed by cropping, random rotation, and flipping. Finally, normalization using the mean was applied to all images.

Random rotation for all the images in all the directions was used for data augmentation. The representation of images after applying various color augmentation operations is displayed in Figure 2. The details of augmentation operations after applying them to the training dataset are given in Table 4. The augmented dataset was 3.6 times larger than the original dataset, and most importantly all DR grades were balanced.

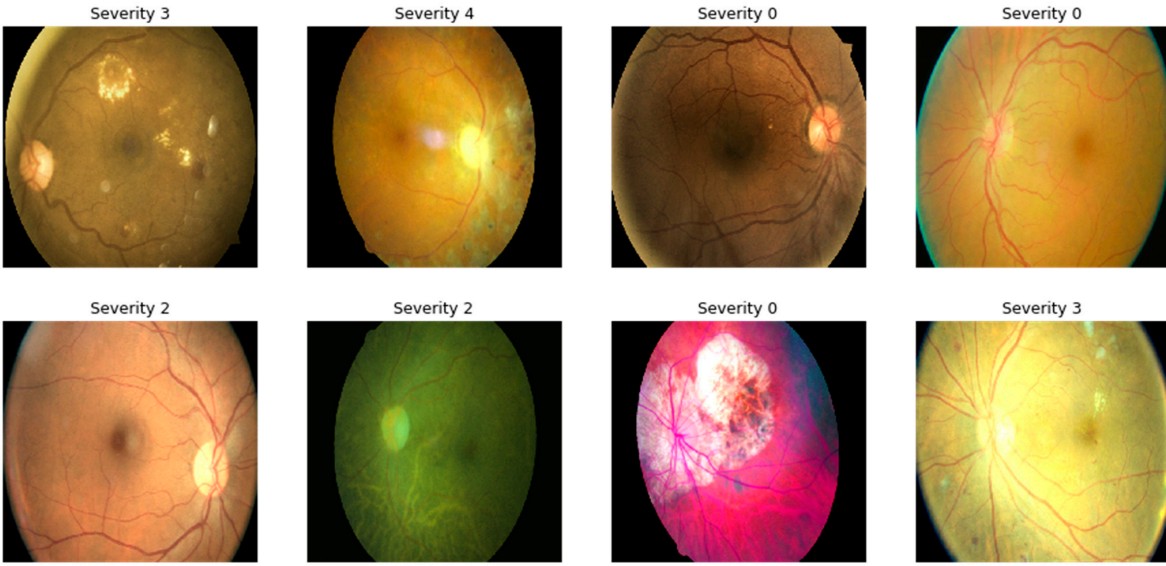

**Figure 2.** Examples of augmented images.

**Table 4.** Distribution for multiclass classification before and after data augmentation.

| Label | Class | Number of Samples | Augmented Samples |
|:---:|:---:|:---:|:---:|
| 0 | Normal | 25,810 | 25,810 |
| 1 | Mild NPDR | 2443 | 24,410 |
| 2 | Moderate NPDR | 5292 | 25,330 |
| 3 | Sever NPDR | 873 | 26,470 |
| 4 | Proliferative DR | 708 | 25,480 |
| Total | | 35,126 | 127,500 |

### 3.3. The Framework Design

Combining rules with feature extraction results from convolutional neural network showed high accuracy in [9]. The combined model was reused in this framework by changing the training data and improving and changing the fuzzy rules following the application domain, which was diabetic retinopathy stage identification. Convolutional layers and pooling layers extracted the most relevant features that were used by rules in the next step; the addition of rules provided a robust stage for identification because it was medical expert-driven and it was domain-specific. Meanwhile, it reduced the training time and provided high accuracy based on the experts' rules.

Figure 3 shows the first part the framework (DRSM-CNNFIS), which is the feature extraction part, with the relevant features obtained by multiple convolutional layers (Layers 1, 2, 3, 5, 6, and 7), and two max pooling layers added (Layers 4 and 8) for dimensionality reduction. The filter size and number were changed and optimized experimentally.

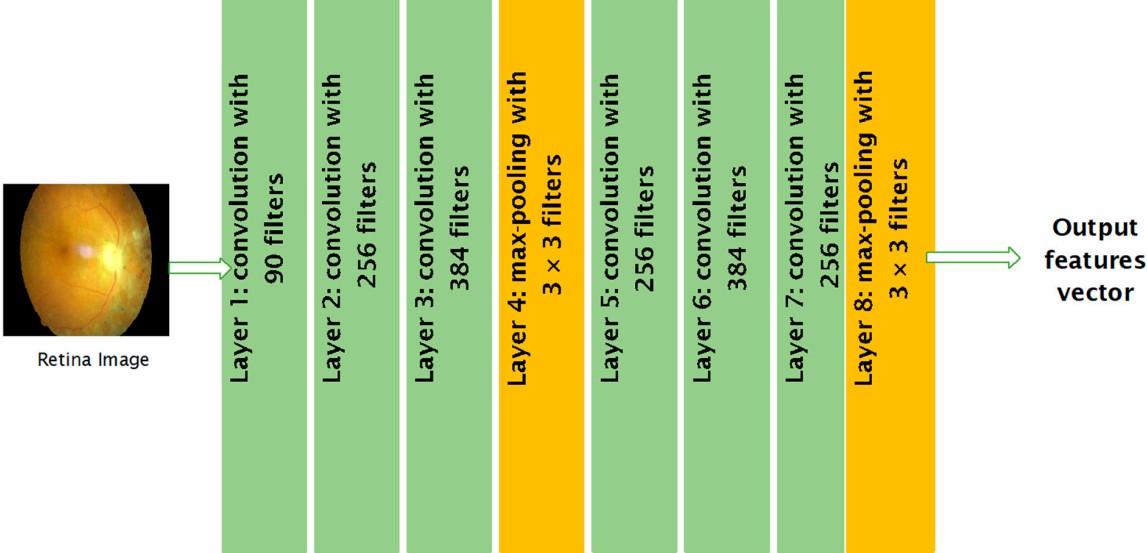

**Figure 3.** Feature extraction part of the DRSM-CNNFIS framework.

The output vector provided the extracted features of the system, which were numeric values used in the diabetic retinopathy stage measurement part, with the fuzzy inference system (FIS) using the constructed rules based on medical experts that explain the direct relation between the extracted vector and the severity stage. The rules were built based on human expert knowledge. The key inputs for the fuzzy inference were features such as microaneurysms, intraretinal hemorrhage, exudates, and macular edema [25]. Linear membership functions were used for the output features: vector mean, standard deviation, max, and min.

Figure 4 shows the second part the framework (DRSM-CNNFIS), which builds the rule-based fuzzy inference system starting with the linear membership functions for the

inputs, and manually added rules based on expert knowledge until the final diabetic retinopathy stage grading.

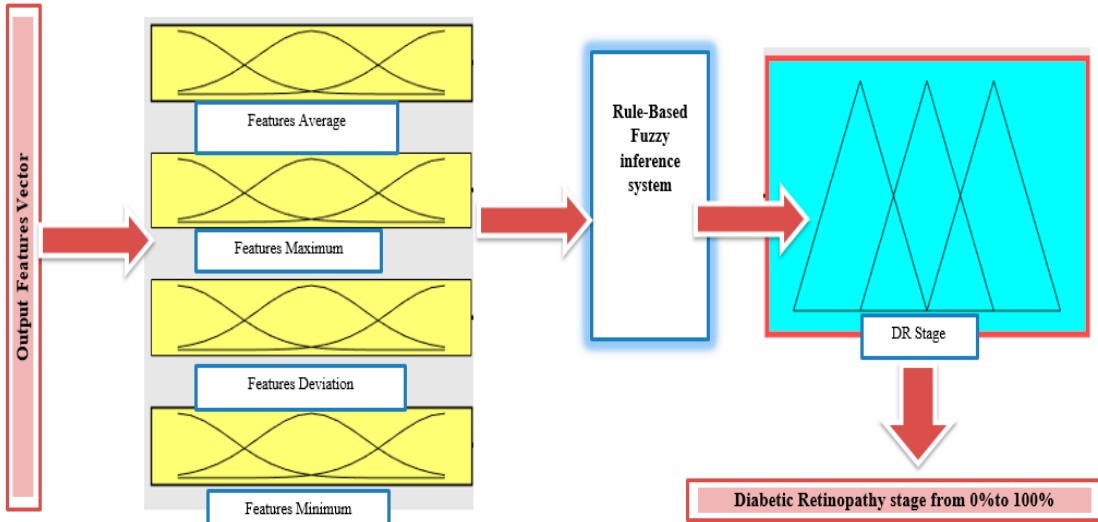

**Figure 4.** Fuzzy rule-based inference system of the DRSM-CNNFIS framework.

*3.4. DRSM-CNNFIS Implementation*

Multiple rounds of convolution and pooling layers were used to provide single-vector output that was used by experts to extract rules associated with each class of diabetic retinopathy (normal, mild, moderate, severe, and proliferative diabetic retinopathy). In the training of CNN with Stochastic, the gradient descent with momentum (SGDM) learning rate was set to 0.1, and an early stop mechanism was used. Tuning for the linear membership function used for the FIS was based on the labeled data. The implementation used Keras and the number of epochs was set to 400. The key inputs for the fuzzy inference were features that could be measured by experts, such as:

- Microaneurysms, which are red patterns that increase the mean value of the output numeric vector;
- Intraretinal hemorrhages, which are outlined patterns starting as dot shape that then defuse into a flame shape and increase the maximum value of the output vector;
- Exudates, which have a yellow or white thick texture and affect the range of the standard deviation of the output vector;
- Macular edema, which occurs when blood leaks into scattered parts of the retina, affecting the minimum value of the extracted features vector.

Four trapezoidal membership functions were used for the output features: the vector mean, standard deviation, max, min, and feature map max. Table 5 presents the description for the linguistics terms used in the membership function for all variables.

**Table 5.** Input and output linguistic variables and their ranges.

| Diabetic Retinopathy Stage Measurement Inference System | | |
|---|---|---|
| **Linguistic Variable** | **Linguistic Value** | **Numerical Range** |
| Input 1: output vector average—microaneurysms | Low, average, high | 0–100 |
| Input 2: output vector maximum—intraretinal hemorrhage | Low, average, high | 0–100 |
| Input 3: output vector standard deviation—exudates | Low, average, high | 0–100 |
| Input 4: output vector minimum—macular edema | Low, average, high | 0–100 |
| Output: diabetic retinopathy (DR) stage | Normal DR (NDR), mild DR (MDR), moderate (MoDR), severe DR (SDR), proliferative DR (PDR) | 0–100 |

To accomplish the stage identification task, medical experts provided rules based on the output feature vector and the associated labels. The medical experts' decisions were based mainly on features, such as microaneurysms, hemorrhages, exudates, and macular edema [35]. Some of the rules used in the fuzzy system are shown in Figure 5.

If output vector average is high and output vector standard deviation is high then the DR is PDR.

If output vector average is high and Output vector standard deviation is low then the DR is NDR.

If output vector average is high and Output vector maximum is high then the DR is MDR.

If output vector average is high and Output vector minimum is high then the DR is PDR.

If output vector average is low and Output vector standard deviation is low then the DR is NDR.

If output vector average is high or output vector maximum is high then the DR is MDR.

**Figure 5.** Fuzzy rules developed by the human expert sample.

The aggregation method used for the rules evaluation is Mamdani inference [9]. Figure 6 shows the membership function for all the linguistic values for the fuzzy output variable: diabetic retinopathy (DR) stage, starting with normal DR (NDR), followed by mild DR (MDR), moderate (MoDR), and severe DR (SDR), and ending with proliferative DR (PDR).

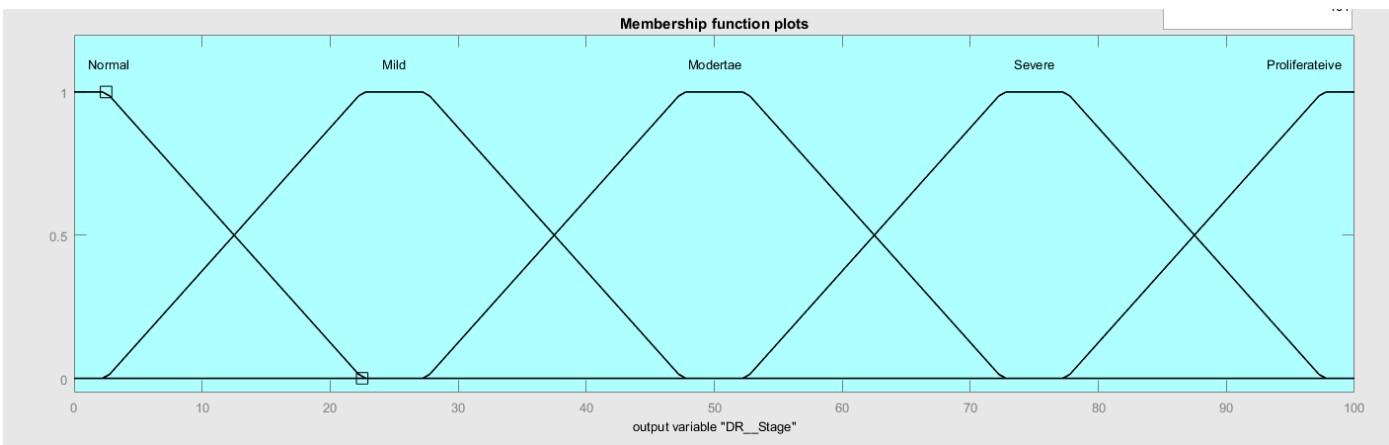

**Figure 6.** The membership functions for the diabetic retinopathy (DR) stage output variable.

After evaluating the rules, the output showed as stage percentage. In this phase, human adjustment of the membership function interval was calculated several times to improve the classification accuracy.

**4. Experimental Results**

The experiments were implemented on two Nvidia Quadro RTX 8000 GPUs in an Ubuntu environment. Fold validation was implemented to obtain more robust results, and due to the size of the dataset, we used five-fold cross-validation to train on 80% of the dataset and tested using 20% of the original dataset during each trial. Early stopping callback was used to minimize validation loss.

Models were evaluated using accuracy precision recall and F1 score; the equations used were applied for the performance assessment mentioned in [36–38].

Table 6 shows that DRSM-CNNFIS outperforms the CNN-only models—DenseNet-201, Inception-ResNet-V2, Inception-V3, ResNet-50, Xception, Majority Vote Ensemble, and Average Ensemble—with an accuracy of 0.9281. The model was evaluated on the macro average, the weighted average for precision, recall, and F1-score to obtain the performance of the model using the single-label classification method of the study. The macro average and the weighted average for precision, recall, and F1-score were evaluated for five classes. Macro averages of 0.7142, 0.7753, and 0.7301, and weighted averages of 0.9371, 0.9281, and 0.9296 were recorded for precision, recall, and F1-score, respectively.

**Table 6.** Evaluation of macro averages and weighted averages for precision, recall, and F1-score and comparative classification results using data with 80% training and 20% testing (validated using five-fold cross-validation) for the diabetic retinopathy classification system.

| Experiment | Accuracy | Macro Average Precision | Macro Average Recall | Macro Average F1-Score | Weighted Average Precision | Weighted Average Recall | Weighted Average F1-Score |
|---|---|---|---|---|---|---|---|
| DenseNet-201 | 0.8226 | 0.5842 | 0.6333 | 0.6021 | 0.8297 | 0.8226 | 0.8248 |
| Inception-ResNet-V2 | 0.8114 | 0.5894 | 0.6487 | 0.6047 | 0.8405 | 0.8114 | 0.8214 |
| Inception-V3 | 0.8050 | 0.5568 | 0.6315 | 0.5676 | 0.8327 | 0.8050 | 0.8101 |
| ResNet-50 | 0.8054 | 0.5662 | 0.5884 | 0.5615 | 0.8150 | 0.8054 | 0.8025 |
| Xception | 0.8122 | 0.5677 | 0.6387 | 0.5948 | 0.8270 | 0.8122 | 0.8183 |
| Majority Vote Ensemble | 0.8378 | 0.6108 | 0.6379 | 0.6137 | 0.8437 | 0.8378 | 0.8370 |
| Average Predictions Ensemble | 0.8381 | 0.6080 | 0.6522 | 0.6200 | 0.8471 | 0.8381 | 0.8396 |
| DRSM-CNNFIS | 0.9281 | 0.7142 | 0.7753 | 0.7301 | 0.9371 | 0.9281 | 0.9296 |

Tables 7–13 illustrates the performance of the five models and the ensemble methods in the class-specific classification in terms of precision, recall, and F1 score. Our framework (DRSM-CNNFIS) shows robust behavior in detecting classes (0, 1, 2, 3, and 4), as shown in Table 14, compared with the distorted behavior for other models that have different performances in each class (0, 1, 2, 3, and 4).

**Table 7.** Experiment 1—DenseNet-201 CNN model class-specific metrics.

| Classes | Precision | Recall | F1-Score |
|---|---|---|---|
| Class 0—No Diabetic Retinopathy | 0.9256 | 0.9209 | 0.9232 |
| Class 1—Mild | 0.3615 | 0.4053 | 0.3821 |
| Class 2—Moderate | 0.6659 | 0.5699 | 0.6142 |
| Class 3—Severe | 0.4296 | 0.6727 | 0.5244 |
| Class 4—Proliferative Diabetic Retinopathy | 0.5385 | 0.5978 | 0.5666 |

**Table 8.** Experiment 2—Inception-ResNet-V2 CNN model class-specific metrics.

| Classes | Precision | Recall | F1-Score |
|---|---|---|---|
| Class 0—No Diabetic Retinopathy | 0.9350 | 0.9014 | 0.9179 |
| Class 1—Mild | 0.3149 | 0.5199 | 0.3922 |
| Class 2—Moderate | 0.7129 | 0.5370 | 0.6126 |
| Class 3—Severe | 0.3921 | 0.6568 | 0.4911 |
| Class 4—Proliferative Diabetic Retinopathy | 0.5922 | 0.6281 | 0.6096 |

**Table 9.** Experiment 3—Inception-V3 CNN model class-specific metrics.

| Classes | Precision | Recall | F1-Score |
|---|---|---|---|
| Class 0—No Diabetic Retinopathy | 0.9260 | 0.9166 | 0.9213 |
| Class 1—Mild | 0.3167 | 0.4751 | 0.3801 |
| Class 2—Moderate | 0.7303 | 0.4254 | 0.5376 |
| Class 3—Severe | 0.3453 | 0.7205 | 0.4669 |
| Class 4—Proliferative Diabetic Retinopathy | 0.4658 | 0.6198 | 0.5319 |

**Table 10.** Experiment 4—ResNet-50 CNN model class-specific metrics.

| Classes | Precision | Recall | F1-Score |
|---|---|---|---|
| Class 0—No Diabetic Retinopathy | 0.8997 | 0.9327 | 0.9159 |
| Class 1—Mild | 0.2768 | 0.3480 | 0.3083 |
| Class 2—Moderate | 0.7467 | 0.4215 | 0.5388 |
| Class 3—Severe | 0.4124 | 0.6364 | 0.5004 |
| Class 4—Proliferative Diabetic Retinopathy | 0.4955 | 0.6033 | 0.5441 |

**Table 11.** Experiment 5—Xception CNN model class-specific metrics.

| Classes | Precision | Recall | F1-Score |
|---|---|---|---|
| Class 0—No Diabetic Retinopathy | 0.9326 | 0.8990 | 0.9155 |
| Class 1—Mild | 0.3628 | 0.4061 | 0.3832 |
| Class 2—Moderate | 0.6177 | 0.6083 | 0.6130 |
| Class 3—Severe | 0.3914 | 0.6795 | 0.4967 |
| Class 4—Proliferative Diabetic Retinopathy | 0.5343 | 0.6006 | 0.5655 |

**Table 12.** Experiment 6—Average Predictions Ensemble class-specific metrics.

| Classes | Precision | Recall | F1-Score |
|---|---|---|---|
| Class 0—No Diabetic Retinopathy | 0.9312 | 0.9379 | 0.9345 |
| Class 1—Mild | 0.3983 | 0.4701 | 0.4312 |
| Class 2—Moderate | 0.7405 | 0.5554 | 0.6347 |
| Class 3—Severe | 0.4080 | 0.6750 | 0.5086 |
| Class 4—Proliferative Diabetic Retinopathy | 0.5622 | 0.6226 | 0.5908 |

**Table 13.** Experiment 7—Majority Vote Ensemble class-specific metrics.

| Classes | Precision | Recall | F1-Score |
|---|---|---|---|
| Class 0—No Diabetic Retinopathy | 0.9241 | 0.9458 | 0.9348 |
| Class 1—Mild | 0.3898 | 0.4377 | 0.4124 |
| Class 2—Moderate | 0.7563 | 0.5311 | 0.6240 |
| Class 3—Severe | 0.4053 | 0.6659 | 0.5039 |
| Class 4—Proliferative Diabetic Retinopathy | 0.5785 | 0.6088 | 0.5933 |

**Table 14.** Experiment 8—DRSM-CNNFIS class-specific metrics.

| Classes | Precision | Recall | F1-Score |
|---|---|---|---|
| Class 0—No Diabetic Retinopathy | 0.9871 | 0.9685 | 0.9777 |
| Class 1—Mild | 0.7796 | 0.8754 | 0.8247 |
| Class 2—Moderate | 0.8663 | 0.8711 | 0.8687 |
| Class 3—Severe | 0.7896 | 0.8854 | 0.8348 |
| Class 4—Proliferative Diabetic Retinopathy | 0.7785 | 0.8088 | 0.7934 |

## 5. Conclusions and Future Work

In this hybrid framework (DRSM-CNNFIS), the diabetic retinopathy stage identification process provided robust performance in all classes (normal DR (NDR), mild DR (MDR), moderate (MoDR), severe DR (SDR), and proliferative DR (PDR)), with an overall accuracy of 93%. Membership functions were constructed and tuned based on labelled data [14] that increased only by 3.6-fold compared to the original data. In the first step, the convolutional neural network was used to obtain the output vector of image features. In the next step, a fuzzy rule-based system was implemented based on human expert knowledge to measure the stage of DR. The final framework showed better performance compared with existing CNN models and ensemble models mentioned in the related work: DenseNet-201, Inception-ResNet-V2, Inception-V3, ResNet-50, Xception, Majority Vote Ensemble, and Average Ensemble. The proposed system showed better results and robust performance for multiclass classification with a weighted average of 0.9371, 0.9281, and 0.9296 for precision, recall, and D1-score, respectively, using the five-fold cross-validation method. In the future, automated method for rule extraction in the fuzzy rule-based system based on the training data will be implemented.

Implementing the proposed model in clinical practice at hospitals and ophthalmology offices will enable regular automated diagnostic measurement, which will save time and cost, and increase the chance of early stage diagnosis, as the results of the proposed model provide stable accuracy for all classes. Furthermore, dealing with noisy or low-quality data can open the door for the future of using mobile phone cameras for such applications.

The proposed model overcomes data dependency problems for deep learning models by using human expert (ophthalmology consultant) knowledge, but the limitation of this work can be seen in different aspects, such as the variation of the performance using different devices and the ethical aspects of health application automation.

**Funding:** This research received no external funding.

**Institutional Review Board Statement:** Not applicable.

**Informed Consent Statement:** Not applicable.

**Conflicts of Interest:** The author declares no conflict of interest.

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
