# Peer review of "Hybrid Framework for Diabetic Retinopathy Stage Measurement Using Convolutional Neural Network and a Fuzzy Rules Inference System"

_asi, doi:10.3390/asi5050102_

Round 1

Reviewer 1 Report

I thank the author for the hard work on the subject.  Overall the concept of this scientific paper and the modeling make logical sense and this is important work.  That being said, I think there are some improvements that must be made prior to publication.  An overall review of English editing and grammar must be done; if the author has trouble with this, I recommend a professional service as there are many grammatical errors throughout the paper that must be improved upon which would be better for reader viewing.  I list some suggestions at the end.  Besides that, the graphics and tables seem clear and make logical sense and the results section runs smoothly.  Conclusions are straightforward and to the point; I do not know if you want to add any significant meaning to what these results would impact in a clinical practice for hospitals and or ophthalmology offices.  That may make some stronger impact with the conclusions of this work.

Specific points to correct:

Lines 27 through 28 need to be written due to bad grammar

In line 31 it should be “tools”

Line 30 age should read “a difficult”

Lines 41 through 45 should be rewritten for better flow

Line 51 you should drop the word “therefore”

I do not think you need lines 54 through 57; instead I would state the goal of your research as opposed to listing out the parts of the paper.  I think your goal with this machine learning on retinal scanning is important and should be written out here as that is what will impact readers

Line 60 is not a sentence, rewrite it

Dropped the capitalization in line 64

Line 74 should read “extension of “

Dropped the word “many” in line 77

In lines like number 96, best state the authors (lead author et al) as opposed to just using the word authors

In line 99 it should be “adds “

Inline 159 drop the capitalization in the middle the of the sentence

These are just some of the examples I found along the way, the rest of the papers better but needs an overall English grammar over hall

Reviewer 2 Report

Paper deals with important task. The authors proposed a hybrid intelligent framework of conventional neural network and fuzzy inference system to measure the stages of DR automatically, Diabetic Retinopathy Stage Measurement using Conventional Neural Network and Fuzzy inference system (DRSM-CNNFIS)..

Suggestions:

1.       The introduction section should be extended using more clearly the motivation of this paper.

2.       It would be good to add point-by-point the main contributions at the end of the Introduction section

3.       It would be good to add the remainder of this paper

4.       The related works section should be extended using non-iterative approaches for solving the stated task. Authors should use this paper: DOI: 10.1007/978-3-030-82014-5_47

5.       The authors should add all optimal parameters for all investigated methods

6.       The conclusion section should be extended using: 1) limitations of the proposed approach; 2) prospects for future research.

Round 2

Reviewer 1 Report

I thank the authors for this revised manuscript. All corrections and suggestions have been integrated into the new revision and it reads much better, has better flow, and provides a better explanation of importance for readers of the journal. No further suggestions at this time, thank you for your hard work and making this a much improved and stronger manuscript.